# Face-to-Face Assembly of Ag Nanoplates on Filter Papers for Pesticide Detection by Surface-Enhanced Raman Spectroscopy

**DOI:** 10.3390/nano12091398

**Published:** 2022-04-19

**Authors:** Sulin Jiao, Yixin Liu, Shenli Wang, Shuo Wang, Fengying Ma, Huiyu Yuan, Haibo Zhou, Guangchao Zheng, Yuan Zhang, Kun Dai, Chuntai Liu

**Affiliations:** 1School of Materials Science and Engineering, Key Laboratory of Materials Processing and Mold (Zhengzhou University), Ministry of Education, Zhengzhou 450001, China; jzr1771516@163.com (S.J.); 18860365730@163.com (S.W.); ctliu@zzu.edu.cn (C.L.); 2Henan Key Laboratory of Advanced Nylon Materials and Application (Zhengzhou University), Zhengzhou University, Zhengzhou 450001, China; 3Key Laboratory of Material Physics, School of Physics and Microelectronics, Ministry of Education, Zhengzhou University, Zhengzhou 450001, China; lyx_zzu@163.com (Y.L.); mafy@163.com (F.M.); 4School of Food Science and Engineering, Henan University of Technology, Lianhua Road 100, Zhengzhou 450001, China; wangshenli@126.com; 5Henan Key Laboratory of High Temperature Functional Ceramics, School of Materials Science and Engineering, Zhengzhou University, Zhengzhou 450001, China; hyyuan@zzu.edu.cn; 6Institute of Pharmaceutical Analysis, College of Pharmacy, Jinan University, Guangzhou 510632, China

**Keywords:** SERS 1, face-to-face assembly 2, Ag nanoplates 3, chemical sensing 4

## Abstract

Surface-enhanced Raman spectroscopy (SERS) technology has been regarded as a most efficient and sensitive strategy for the detection of pollutants at ultra-low concentrations. Fabrication of SERS substrates is of key importance in obtaining the homogeneous and sensitive SERS signals. Cellulose filter papers loaded with plasmonic metal NPs are well known as cost-effective and efficient paper-based SERS substrates. In this manuscript, face-to-face assembly of silver nanoplates via solvent-evaporation strategies on the cellulose filter papers has been developed for the SERS substrates. Furthermore, these developed paper-based SERS substrates are utilized for the ultra-sensitive detection of the rhodamine 6G dye and thiram pesticides. Our theoretical studies reveal the creation of high density hotspots, with a huge localized and enhanced electromagnetic field, near the corners of the assembled structures, which justifies the ultrasensitive SERS signal in the fabricated paper-based SERS platform. This work provides an excellent paper-based SERS substrate for practical applications, and one which can also be beneficial to human health and environmental safety.

## 1. Introduction

Surface-enhanced Raman spectroscopy (SERS) is one promising surface precautionary technology for the selective and sensitive detection of analytic molecules in the field of human health and environmental safety [1]. Even an ultra-low concentration of target analytes can be harmful or deadly for the environment and living system. Therefore, the development of stable, portable, and sensitive technology is urgently required for the fingerprint identification of target molecules. The non-destructive, rapid, and label-free characteristics of SERS make it to well-suited to meeting this practical need. One cornerstone of SERS is its ability to probe Raman scattering of molecules nearby or on the surface of plasmonic metal nanoparticles (NPs). The well-reorganized chemical and physical enhancements in SERS are due to the mechanisms of charge transfer and localized electromagnetic field (EF) enhancement, respectively [2,3,4,5].

The unique morphology features of plasmonic metal NPs exhibit excellent localized surface plasmon resonance (LSPR) properties, especially in anisotropic metal NPs [6,7,8,9,10] or nanoparticle dimers [11,12,13]. Often, the extreme enhanced localized EF is observed on the high-index facets, sharp edges, and tips of plasmonic metal NPs, leading to the so-called hotspots. Furthermore, the assembled nanostructures also create hotspots between constitute NPs, which normally feature more enhanced localized EF within well-controlled nanogaps, due to the formation of the hybrid plasmonic response. Thus, assembly engineering of plasmonic metal NPs for a large yield of hotspots is an efficient strategy for trace-amount SERS detection [14,15,16,17]. A variety of chemical and physical methods are now available for large-scale fabrication of assembled metal nanostructures. The complex types of assembled metal nanostructures possessing strongly enhanced EF have become a promising platform in the fields of bio/chemical sensing, photocatalysis, cancer therapy, and photonics [18,19,20,21].

However, cost-effective and high efficiency SERS substrates are still necessary for obtaining the single-molecular SERS signal, as well as for other applications [22]. Cellulose filter papers combined with plasmonic metal NPs have been applied as SERS substrates because of their low-cost, easy fabrication, high surface area and porosity, mechanical strength, environmentally friendly material and biodegradability, easy recovery and reusability, as well as strong adhesion to a variety of materials [23,24,25,26,27]. The paper-based flexible SERS substrates have gained tremendous use in research due to the advantages of inexpensive fabrication procedures, flexibility, and the ease of tracking analysts in sample detection [28]. Recently, the development and applications of paper-based SERS substrates has received incredible attention in the field of the detection of hazardous materials (a summarization of previous paper-based SERS sensors is shown in Appendix A). Additionally, a series of paper-based SERS substrates has been developed for thiram detection (Appendix A) [29,30,31], such as Ag NPs on cellulose filter papers, Au NPs coated cellulose fiber, and core-shell Ag@SiO_2_ NPs coated filter paper. These substrates exhibit sensitive detection of thiram, and their LODs are 4.6261 ng/cm^2^, 1 nM, and 1 nM, respectively. We have previously developed a general fabrication methodology for cellulose filter paper nanocomposites for SERS sensing and chemical catalysis [32,33]. During the preparation approaches, filter papers nanocomposites were dried using a hair dryer after dipping into the metal NPs’ organic ink. However, most of synthetic plasmonic metal NPs are in the aqueous solution [27]. The transfer routes of water-soluble metal NPs into organic solution are complex and time-consuming. In previous research, hotspots on the SERS substrates can mediate the electromagnetic field coupling. For instance, when an ordered micropyramid array of Ag NPs was developed, the EF enhancement reached 180 [34]. Zhu et al. prepared an Ag nanocubes/graphene-oxide/Au-nanoparticles composite film, and the maximum value of localized electric field between the Ag-NCs and Au nanospheres reached 200 [35]. In order to overcome the electrostatic repulsion, they demonstrated a chloride ion-assisted self-assembly method to deposit Ag NPs on filter paper. The enhancement factor of the substrate was higher than 6.4 × 10^5^, with a limit of detection of 1 × 10^−8^ M for 4-mercaptobenzoic acid [36]. The controllable assembly of metal NPs has not yet been achieved for the optimization of the SERS signal. The solvent-evaporation induced assembly of metal NPs is well known for the simple fabrication of a large-scale packed density of metal NPs [37,38].

In the present manuscript, we propose one strategy that aims to utilize solvent-evaporation mediated assembly of metal NPs on the cellulose filters, in which an excellent paper-based SERS platform with large controlled hotspots toward chemical sensing have been created. A highly concentrated solution of silver nanoplates ink was firstly collected. Afterwards, a simple construction methodology of self-assembled Ag nanoplate films on cellulose filter papers were developed. The theoretical simulations have been carried out to verify the high density of hotspots with huge localized EF near the corners of the assembled structures. As a result, a cost-effective and highly efficient paper-based SERS substrate is enabled and experimentally tested for he detection of ultra-low concentration of rhodamine 6G dye and thiram pesticides.

## 2. Materials and Methods

### 2.1. Reagents and Apparatus

The silver nitrate (AgNO_3_), sodium borohydride (NaBH_4_), hexadecyltrimethylammonium bromide (CTAB), L-ascorbic acid (AA), sodium hydroxide (NaOH), and rhodamine 6G (R6G) are purchased from Adamas Beta and are used without further processing. The filter paper was purchased from the Hangzhou Fuyang Paper Company. Milli-Q water was used in all of the experiments. The UV-vis absorption spectrum was determined using the UV-1900 (SHIMADZU). The morphology of the samples was characterized by scanning electron microscopy (SEM, JEOL-6700F), transmission electron microscopy (TEM, JEM-2010, 200 kV), and high-resolution transmission electron microscopy (HRTEM, JEM-2100F, 200 kV). The composition and phase of the products were detected by an X-ray diffractometer (Bruker D8 ADVAVCE). SERS spectra were recorded by using Raman spectroscopy (HORIBA LabRAM HR Evolution) at ambient temperature (633 nm laser; 50 × objective; 10 s Acquisition time).

### 2.2. Synthesis of Silver Nanoplates

Silver nanoplates were synthesized from the solution process with a previously developed strategy [39]. In brief, 5 mL of AgNO_3_ (10 mM) and 0.3 mL of NaBH_4_ (10 mM) were added into 5 mL of CTAB (0.4 mM) solution successively, and the solution was stored in the dark. After one hour, we added 2 mL of AgNO_3_ (10 mM) and AA (0.1 M) into the 39 mL CTAB (10 mM), followed by 250 μL of the prepared seed solution, and 400 μL NaOH (1 M). The resultant solution was then stirred for 10 min. The pure Ag nanoplates were obtained by centrifugation at 8500 rpm for ten minutes and washed three times. Finally, the Ag nanoplates precipitates were redispersed in 2 mL of Milli-Q water for further use.

### 2.3. Preparation of Paper-Based SERS Substrates

The filter paper (1 cm × 1 cm) was slowly dipped into a vial containing Ag nanoplates ink and dried at 40 degrees. The whole process was repeated until the solution ran out.

### 2.4. SERS Measurements

The prepared paper-based SERS substrates were immersed in different concentrations of R6G water solution, thiram acetone solution, or thiram orange juice solution, and then stored overnight. The orange juice solution was prepared by a juicer, followed by being centrifuged at 7000 r/min for ten minutes to remove sediment. SERS measurements were conducted after drying the substrates in the fume hood.

### 2.5. Theoretical Simulation

To study the plasmonic response of the silver triangle nanoplates aggregate in water, we carried out the corresponding simulations by solving Maxwell’s equations for electromagnetic fields with the boundary element method [40,41], as implemented in the MNPBEM toolkit [42]. For these simulations, we utilized the dielectric of silver by Johnson–Christy [36], and that of water by Hale–Querry.

To investigate the origin of peaks in SERS, we carried out density functional theory (DFT) calculations for the R6G cation [8], and the thiram molecule with the NWChem 6.8 package [9],utilizing the B3LYP hybrid functional and 6-31G* basis set. We first optimized the molecular structure, then carried out frequency analysis to confirm the absence of imaginary frequencies. After this, we calculated the Raman spectrum [10,11] for the 633 nm laser excitation.

## 3. Results & Discussion

### 3.1. Morphological and Optical Characterization of Silver Triangle Nanoplates

The fabrication process of silver nanoplates was mentioned in the experimental section. As revealed by transmission electron microscopy (TEM) images (Appendix A), a large number of Ag nanoplates with a size distribution of 52.7 ± 4.6 nm were obtained (Appendix A). Their crystal structure was obtained through high-resolution transition electron microscopy (HRTEM) and reported (Figure 1a,b), indicating a 0.24 nm lattice distance on typical (111) facets of Ag nanoplates. X-ray diffraction (XRD) analysis was performed to further study the crystal structure of Ag nanoplates (Figure 1c). The characteristic peaks at 38°, 45°, 64°, and 77° correspond to the different planes of (111), (200), (220), and (311), respectively. In Figure 1d, UV-vis extinction spectra showed three LSPR bands, corresponding to the in-plane dipolar resonance, and out-of-plane multipolar resonance, respectively. Additionally, the maximum SPR band is localized at 620 nm, which is sensitive to the injected amounts of seeds. It will blue-shift from 620 nm to 556 nm as the volume of seeds changes from 250 μL to 700 μL. To gain insight into the measured extinction spectra of the Ag NPs in water, we carried out the corresponding simulations. Figure 1e reveals that the extinction spectra of Ag NPs varied with length from 35 nm to 70 nm (for a fixed thickness of 14 nm). The extinction spectra showed one maximum LSPR peak at around 600 nm, and several small peaks at around 420 nm, for one nanoplate.

The generation of maximum peaks at ca. 600 nm was the result of two degenerated dipolar modes (Figure 1f), which were formed by positive charges at one corner and negative charges at another corner or other two corners. The small peaks were due to several higher-order plasmon modes, while the two important modes were formed by the alternation of negative and positive charges once along one arm, but twice along another arm (Figure 1g). Furthermore, Figure 1e also showed that as the length of the nanoplates increased from 35 nm to 70 nm, the peak at the longer and shorter wavelengths red-shifted more than 100 nm, and 10 nm, respectively, and their extinction also increased. In comparison to the spectrum variance with the measured extinction spectrum (Figure 1d), we might conclude that the inhomogeneity of the nanoplate size is responsible for the broadening spectrum. The other effects, like the orientations of the nanoplates and the illumination conditions, would contribute to further smoothing the measured absorptions.

### 3.2. Self-Assembly of Silver Nanoplates

The face-to-face assembly of Ag nanoplates on the filter paper was developed by a simple method, as shown in the schematic diagram of the fabrication of paper-based SERS substrates (Figure 2a). At the beginning, the concentrated colloid ink was collected into a vial. Then, a piece of filter paper was dipped into the Ag nanoplates ink and dried at 40 °C. The whole process was repeated several times until the colloid ink ran out. The color change of the filter paper during the whole process proved that a large number of Ag nanoplates were adsorbed on the surface (Appendix A). Compared to scanning electronic microscopy (SEM) images of the pure surface (Appendix A), SEM images in Figure 2b and Appendix A clearly revealed the uniform distribution of the Ag NPs on cellulose filter paper. Face-to-face assembly of Ag nanoplates occurred during the entire dipping processes, because the higher surface energy of the facets drove the information entropy [38,43,44,45,46]. SEM image in Figure 2c clearly showed that Ag nanoplates were stacked and arranged in a face-to-face manner.

In order to investigate the distribution of localized EF around the face-to-face assembled nanostructures, the plasmonic response and the localized EF enhancement of the single silver nanoplate and the assembled nanostructures were simulated, respectively. Initially, we focused on the extinction spectra under the plane-wave illumination with polarization along the *x*-axis (black solid lines), *y*-axis (red dashed lines), and *z*-axis (blue dotted lines). For the single Ag NP, there was a dominant peak at 478 nm under the x- and y-polarized light illumination, as shown in Figure 3a. For the aggregate with two stacked NPs with a 1 nm gap, a peak at around 679 nm appeared under the z-polarized light illumination, as shown in Figure 3b. For the aggregate with one more NP, the peak appeared blue-shifted to around 623 nm with stronger extinction, and there was another peak at around 1100 nm under the light illumination with any polarization, as shown in Figure 3c. When more NPs were added to the aggregate, both peaks experienced blue-shift, and the amount of shift increased with increasing number of NPs, as shown in Figure 3d,e. Furthermore, we had also investigated the extinction spectrum for an aggregate with five silver nanoplates of 70 nm length, as seen in Figure 3f. In this case, the peak with short wavelength red-shifted to the wavelength of 639 nm, and the peak with the longer wavelength was split into two. Considering the 633 nm laser used in our SERS experiment, we expected that the last aggregate was more relevant for our experiment and, thus, we examined the near-field enhancement afforded by this aggregate. After identifying the two relevant plasmonic modes, the aggregate formed by five nano-plates were focused. Here, we saw that the localized EF enhancement was significant inside the gaps and the boundaries of the gaps, which are characteristics of so-called hybrid gap plasmons. Furthermore, the EF enhancement was significant on all the edges and the middle of all sides for the 633 nm excitation (Figure 3g), while EF enhancement was significant on one edge and the middle of one side for the 954 nm excitation (Figure 3h), and on two edges for the 994 nm excitation (Figure 3i). This result indicates that the plasmons at these wavelengths are different kinds of gap plasmons. We noticed that the maximal enhancement was about 10^2^ = 100, 10^2.5^ = 316, and 10^3^ = 1000 for the 633 nm, 954 nm and 994 nm excitation, respectively (see Figure 3b,d,f). Considering the SERS enhancement is the fourth power of the field enhancement [6], we expect a SERS enhancement of ${\left(\frac{E}{E_{0}}\right)}^{4}=10^{8},10^{10},10^{12}$ for the molecules absorbed on the sides and edges of the gaps under 633 nm, 954 nm, and 994 nm excitation. According to the above discussions, assembly of Ag NPs can afford the controlled hotspots with significant Raman enhancement, which might effectively improve the sensitivity of SERS detection.

### 3.3. SERS Measurements of R6G

To investigate the SERS performance of fabricated paper-based SERS substrates, SERS measurements were performed on the substrates with different concentrations of R6G molecules, ranging from 10^−5^ to 10^−13^ M. R6G (10^−5^ M). These concentrations were selected to test the homogeneity of SERS signals. The optical image in Figure 4a and the SERS mapping (for the peak at 613 cm^−1^) in Figure 4b showed a nearly uniform SERS signal distributed on the substrate surface. The SERS spectra of R6G with different concentrations were shown in Figure 4c. The limitation of detection of R6G was quite low at 10^−13^ M, indicating that the SERS substrate had an excellent performance in trace detection. In addition, the Raman intensity at 613 cm^−1^ was selected as an ordinate, and the function between its intensity and different concentrations was plotted in a log scale (Figure 4d), where five measurements were carried out for each concentration (Appendix A). Additionally, we also simulated Raman spectra of R6G molecule with DFT calculation as shown in Appendix A and observed several typical Raman peaks at 613 cm^−1^, 768 cm^−1^, 1181 cm^−1^, 1311 cm^−1^, 1361 cm^−1^, 1511 cm^−1^, and 1650 cm^−1^, which were reported previously in the literature [47]. By comparing Figure 4c and Appendix A, we identified the vibrational modes contributing to the mainly Raman peaks in the measured spectrum (see the blue dashed lines for the vibrational assignment). We noticed that the peaks were different in the measured and calculated spectrum, and several peaks in the calculated spectrum did not appear in the measured spectrum. These differences can be attributed to the different selection rules in the normal Raman and SERS. Furthermore, we showed the pattern of the mainly Raman active modes in Appendix A and identified that the 628.50 cm^−1^ mode corresponds to an in-plane deformation of the xanthene ring. The 822.18 cm^−1^ and 827.69 cm^−1^ mode correspond to the C-H out-of-plane bend, while the 1206.63 cm^−1^ corresponds to the C-H in-plane bend. Additionally, 1327.34 cm^−1^ corresponds to C-C stretching, while the 1398.70 cm^−1^, 1548.51 cm^−1^, 1554.10 cm^−1^, and 1696.71 cm^−1^ correspond to aromatic C-C stretching. The substrates were also tested for durability at room temperature. As shown in Appendix A, the signal intensity decreased with the exposure time. This is mainly due to the oxidation reaction of silver.

### 3.4. SERS Application for Thiram Detection

As a common toxic chemical substance, thiram is widely used in plant pest control [48], and thus it is selected as the probe molecule for residual detection in this study. More precisely, we have used the fabricated paper-based SERS substrates for the measurement of thiram in the food safety. The SERS spectra of thiram with different concentrations as reported in Figure 5a showed that limitation of detection is low to 10^−8^ M. In addition, we could clearly see the trend of the peak intensity at 1377 cm^−1^ (Figure 5b), where five measurements were carried out for each concentration (Appendix A). The measured SERS for the thiram molecule showed several typical Raman peaks at 440 cm^−1^, 562 cm^−1^, 935 cm^−1^, 1143 cm^−1^, 1382 cm^−1^, 1441 cm^−1^ and 1509 cm^−1^, as reported in the literature (Appendix A) [49]. 

We observed the same peaks in the calculated Raman spectrum (Appendix A) as in the measured spectrum except for slightly larger Raman shifts and different relative intensity. The latter difference can be attributed to the influence of the chemical absorption of the thiram molecule on the silver surface [49]. We showed also the pattern of the vibrational modes contributing to the Raman peaks in Appendix A. Based on the above results, the different concentrations of thiram were put into orange juice to create real condition (Appendix A). As shown in Figure 5c,d, the detection range in the orange juice was from 10^−4^ M to 10^−7^ M, and the peak position is consistent with the theoretical calculation result. Compared with thiram in acetone, Figure 5c showed a lower peak density at 1377 cm^−1^, but still with a good recognition. In general, the self-assembled substrate can sensitively detect thiram concentration, where the limitation of detection is far lower than the national standard.

## 4. Conclusions

In summary, we have achieved the ultra-sensitive chemical sensors for the R6G molecules and thiram molecules. A simple fabrication approach for the paper-based SERS substrates has been developed via the controllable face-to-face assembly of Ag nanoplates on filter papers. Our theoretical simulations showed that, high density of hot spots was distributed nearby the assembled nanostructures. Therefore, the paper-based SERS substrates showed efficient and homogeneous distribution of SERS spectra dyes and pesticide. The developed paper-based SERS substrates can be potentially applied in the food safety, environment monitoring, and defense security.

## Figures and Tables

**Figure 1 nanomaterials-12-01398-f001:**
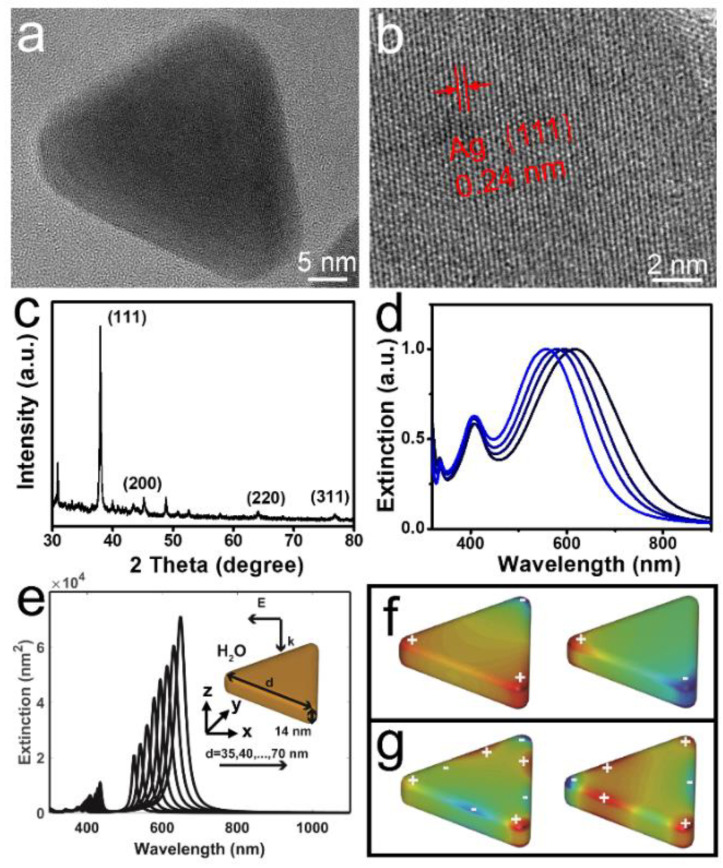
Characterization of silver triangle nanoplates. (**a**) Typical TEM image of single Ag NP. (**b**) HRTEM image of Ag NP. (**c**) XRD of Ag NPs. (**d**) UV-vis extinction spectra of Ag NPs. UV-vis extinction spectra showed three LSPR bands, corresponding to the in-plane dipolar resonance, and out-of-plane multipolar resonance, respectively. Additionally, the maximum SPR band is localized at 620 nm, which is sensitive to the injected amounts of seeds. It will blue-shift from 620 nm to 556 nm as the volume of seeds changes from 250 μL to 700 μL. (**e**) Computed extinction cross-section of Ag NPs (inset of (**a**)) with a thickness of 14 nm and various length in water, which are illuminated by a plane-wave propagating along negative *z*-axis and polarizing along the *x*-axis, as indicated in the inset. (**f**,**g**) The surface charge distribution (blue for negative charge, and red for positive charge) associated with the plasmon peaks at longer (**f**) and shorter wavelengths (**g**), as marked in (**e**), for an Ag nanoplate with a length of d = 53 nm.

**Figure 2 nanomaterials-12-01398-f002:**
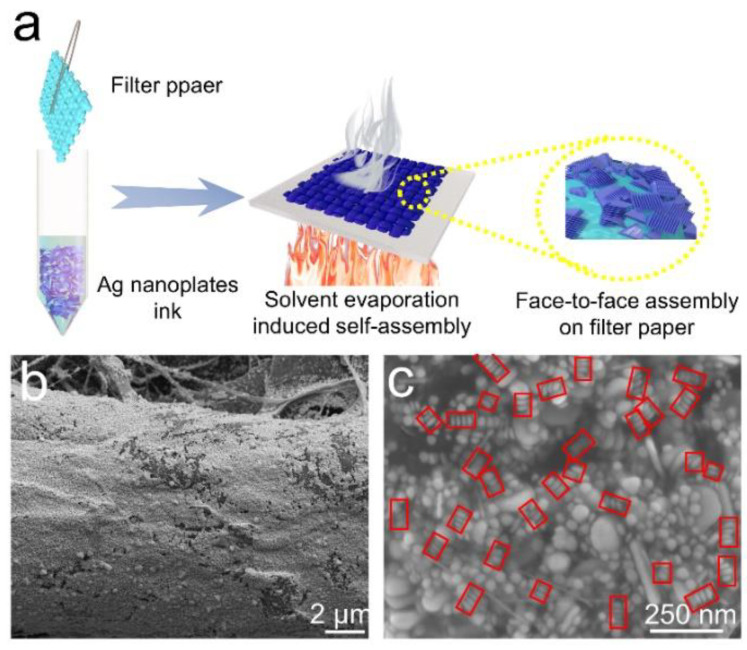
Characterization of paper-based SERS substrates. (**a**) Schematic diagram of fabrication of paper-based SERS substrates. (**b**,**c**) SEM images of Ag NPs coated filter paper. Red boxes in (**c**) indicate the face-to-face stacked silver triangle nanoplates.

**Figure 3 nanomaterials-12-01398-f003:**
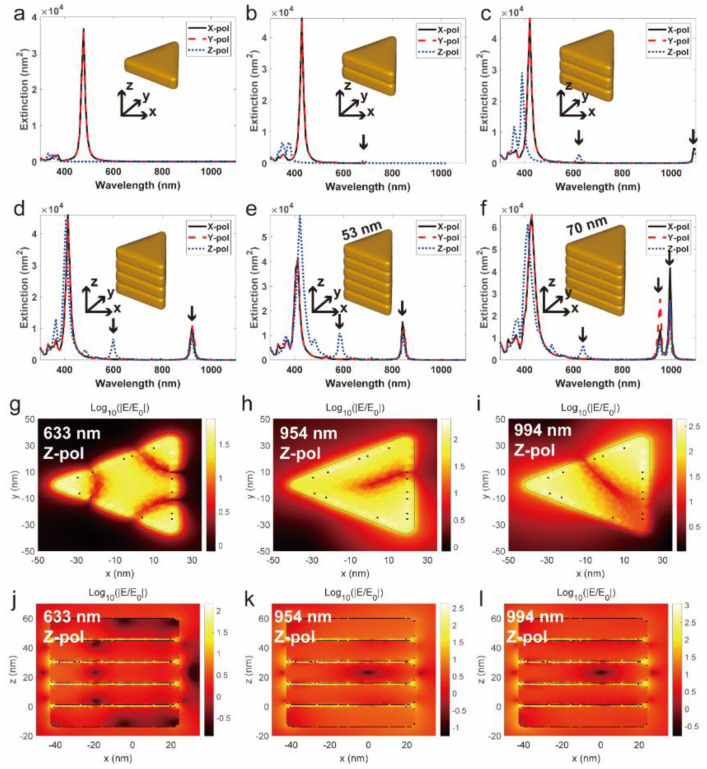
Theoretical simulations of the Ag nanoplates. Computed extinction spectrum for (**a**) single Ag nanoplate and (**b**–**e**) nano-aggregates formed with two (**b**), three (**c**), four (**d**), five (**e**,**f**) silver nanoplates in air under the plane-wave illumination with x-(black solid lines), y-(red dashed lines) and z-polarization (blue dotted lines). The nanoplates have a thickness of 14 nm, a gap of 1 nm and a length of 53 nm (**a**–**e**) or 70 nm (**f**). Computed near-field enhancement in logarithmic scale in the x-y plane (**g**–**i**) and x-z plane (**j**–**l**) through an aggregate with five Ag NPs of 70 nm length under z-polarized light illumination, with the wavelength of 633 nm (**g**,**j**), 954 nm, (**h**,**k**), and 994 nm (**e**,**f**), respectively. The nanoplates have a length of 70 nm and a thickness of 14 nm, and a gap of 1 nm.

**Figure 4 nanomaterials-12-01398-f004:**
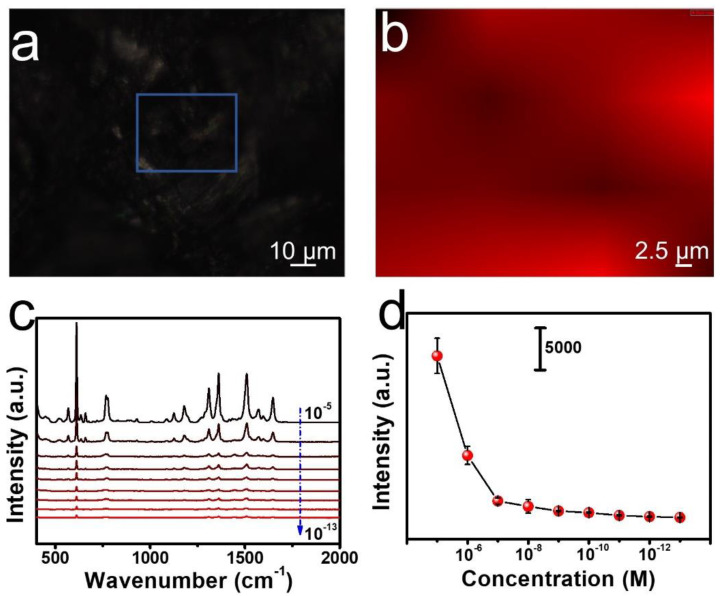
SERS measurements of R6G molecules. (**a**) Optical microscopy image of paper-based SERS substrates. (**b**) Raman mapping image at 613 cm^−1^ with 633 nm laser excitation. (**c**) SERS spectra of different concentration (10^−5^ M to 10^−13^ M) of R6G molecules on substrates under the irradiation of 633 nm laser. (**d**) Plot of SERS intensityfor the paper- based SERS substrates has been developed via the controllable face-to-face assembly of Ag nanoplates on filter paper. Our theoretical simulations showed that a high density of hotspots was distributed nearby the assembled nanostructures. Therefore, the paper-based SERS substrates showed efficient and homogeneous distribution of SERS spectra dyes and pesticide. The developed paper-based SERS substrates can be potentially applied in the food safety, environment monitoring, and defense security industries.

**Figure 5 nanomaterials-12-01398-f005:**
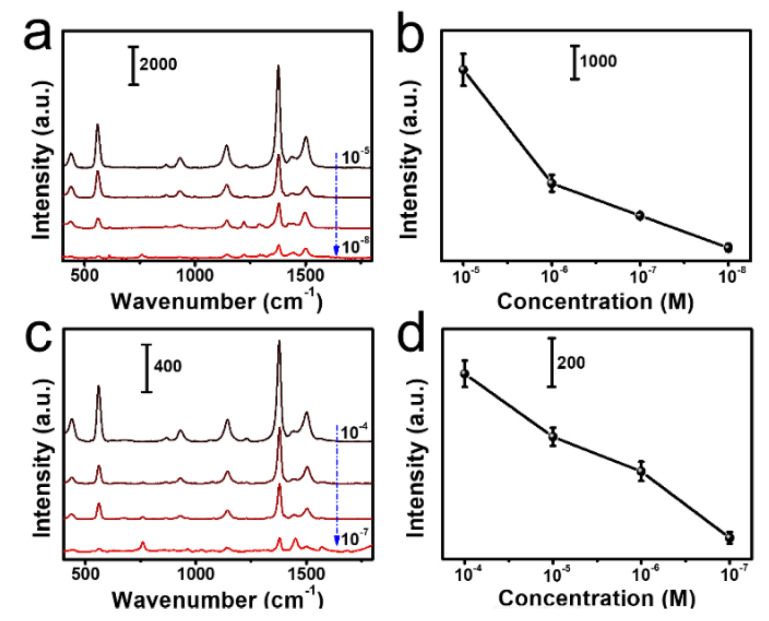
The SERS detection of thiram. (**a**,**c**) SERS spectra of thiram in acetone or orange juice. (**b**,**d**) Plot of concentration-SERS intensity relationship of the 1377 cm^−1^ band.

## Data Availability

Not applicable.

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
