# Peer review of "Face-to-Face Assembly of Ag Nanoplates on Filter Papers for Pesticide Detection by Surface-Enhanced Raman Spectroscopy"

_nanomaterials, 2022, doi:10.3390/nano12091398_

Round 1
Reviewer 1 Report
“Assembly of Ag Nanoplates on Filter Papers for Suface enhanced Raman Spectroscopy” by Sulin Jiao et al
I went through the paper very carefully and thoroughly. Authors studied Assembly of Ag Nanoplates on Filter Papers for Suface enhanced Raman Spectroscopy. In this manuscript, they reported the face-to-face assembly of silver nanoplates via solvent-evaporation strategies on the cellulose filter papers. Furthermore, these developed paper-based SERS substrates are utilized for the ultra-sensitive detection of the Rodamine 6G dyes and thiram pesticides. The present theoretical studies reveal the creation of high density of hotspots with huge localized and enhanced electromagnetic field near the corners of the assembled structures, which justifies the ultrasensitive SERS signal in the fabricated paper-based SERS platform.
1-The paper contains interesting sciences. The impact of the paper is going be good. Also, the quality of the research work presented in the paper is also good. .
2-In general, ideas are well explained and understandable but, some tenses, linkers and grammar structures must be checked.
3-The authors should give the thickness and number of layers of all layers that they calculated. Are these parameters obtained from an optimization process?
- Authors should obtained the novelty of this manuscript compared to published results?
- The authors should argue about the relevance of the temperature dependence of the coating.
- The Introduction does not provide sufficient background. The introduction does not explain the major contributions and novelty of this work. The significance of the proposed solution has not been summed up.
- Is it possible to have results without a theoretical part represented by explanatory equations?
8- The constructive discussions are missing. As mentioned earlier, authors must make a comparative analysis with other similar solutions and back up their claims on how the proposed solution can be considered as high performing compared to others
9- How their results will be affected if they include energy loss in layers.
10- The novelty of this work should be stated explicitly in the text of the manuscript so that readers can get it easily.
11- Authors should compare their results with the published data and different results.
12- There a lot of published papers in this field, authors should be explained the new in these results in an ultra-sensitive chemical sensor.
13- Authors mentioned “As a common toxic chemical substance, thiram is widely used in plant pest control, and thus it is selected as the probe molecule for residual detection in this study” without any evidence
14- Authors should be explained the distribution of electric fields with this structure as well as the equations related. It is will be very useful.
15- Authors should explain one or two application to their work.
16- All figures, symbols, equations should be improved.
17- It seems the title need revision by authors to become more informative.
18- How this device can be stable with these kinds of materials.
19- The whole concept is now unclear to a reader what is the actual effect of the sensitivity of this device?
20- Are every term and structure in the proposed design should be clearly and correctly presented not to mislead the reader.
21-It seems all experimental results need revision and should compare with published data.
22- What do you mean “Suface” in your title?
23-Finally, I recommend that the paper should be revised taking care of the above comments.
I wish to resend this paper after corrections and revise my comments
Reviewer 2 Report
This manuscript presents a paper-based SERS substrate loaded with assembled Ag nanoparticles for Rhodamine and pesticide detection. The data presented in the manuscript appear complete and should be attractive to the readership of Nanomaterials. However, I have some reservation in recommending it for publication as of the current form for the two main reasons. First, the introduction lacks in the recent review of the paper-based SERS detection. One of the authors published the related work back in 2015, but since then, there have been many papers published in the field. See the detailed comments below. The motivation of the work and justification statement should be revised after reflecting on these related papers. Second, there have been too many typos and grammatical errors. Notably, the title of the manuscript has a typo (Suface should be Surface), representing the lack of careful proofreading. Below are some of the comments that may strengthen the quality of the manuscript.
- As stated above, the recent work on filter papers as SERS substrates should be reviewed and used as a basis for what unique contribution this manuscript brings to the research community. Several papers I found from a quick literature review include (all on or after 2015)
- Hasi et al., Chloride ion-assisted self-assembly of silver nanoparticles on filter paper as SERS substrate, Applied Physics A: Materials Science and Processing 118(3), 2015, pp. 799-807
- He et al., Optimizing the SERS enhancement of a facile gold nanostar immobilized paper-based SERS substrate, RSC Advances, 7(27), 2017, pp. 16264-16272
- Zhang et al., Cysteamine-Assisted Highly Sensitive Detection of Bisphenol A in Water Samples by Surface-Enhanced Raman Spectroscopy with Ag Nanoparticle-Modified Filter Paper as Substrate, Food Analytical Methods, 10(6), 2017, pp. 1940-1947
- Zhu et al., Highly sensitive and label-free determination of thiram residue using surface-enhanced Raman spectroscopy (SERS) coupled with paper-based microfluidics, Analytical Methods, 9(43), 2017, pp. 6186-6193
- Wang et al., A “drop-wipe-test” SERS method for rapid detection of pesticide residues in fruits, Journal of Raman Spectroscopy, 49(3), 2018, pp. 493-498
- Moram et al., Ag/Au Nanoparticle-Loaded Paper-Based Versatile Surface-Enhanced Raman Spectroscopy Substrates for Multiple Explosives Detection, ACS Omega, 3(7), 2018, pp. 8190-8201
- Sun et al., Performance enhancement of paper-based SERS chips by shell-isolated nanoparticle-enhanced Raman spectroscopy, Journal of Materials Science and Technology, 35(10), 2019, pp. 2207-2212
- Tegegne et al., Ag nanocubes decorated 1T-MoS2 nanosheets SERS substrate for reliable and ultrasensitive detection of pesticides, Applied Materials Today, 21, 2020, 100871
- Pagano et al., Ag nanodisks decorated filter paper as a SERS platform for nanomolar tetracycline detection, Colloids and Surfaces A: Physicochemical and Engineering Aspects, 624, 2021, 126787
- Ge et al., Detection of Formaldehyde by Surface-Enhanced Raman Spectroscopy Based on PbBiO2Br/Au4Ag4Nanospheres, ACS Applied Nano Materials, 4(10), 2021, pp.10218-10227
- Of course, by no means, this is a complete list of the relevant papers. The authors don’t need to cite all of the suggested papers, but they should clearly state how the current work differs from the past work and what aspect this work improves the paper-based SERS detection.
- What is the maker or model of the filter paper used?
- In Section 2.5, the figures numbers like Figure 2 and 5 are mentioned before Figure 1. This section should be written without referencing the figures.
- Section 3.2. Line 194-195: The authors cited [31, 37, 78] to support the face-to-face self-assembly of Ag nanoplates. But [31] discusses the edge-to-edge assembly of the Ag nanoplates, and the other two didn’t discuss the face-to-face assembly either. Please cite the relevant papers. Another thing. What does “information entropy” have anything to do with self-assembly?
- Figure 6f: This figure is very hard to read.
- As mentioned above, the authors need to carefully proofread the manuscript for errors.
Reviewer 3 Report
The paper was well written and interesting. I have only one minor comments.
Please compare the obtained results with similar approach such as plasmonic paper. It would enhance the value of the paper.
Round 2
Reviewer 1 Report
Now the updated version is ok and authors considered all comments